# FactNLI: Dynamic and Automated Fact-Based Augmentation of NLI Benchmarks

## Abstract

Natural Language Inference (NLI) is a core task for language understanding, yet existing NLI datasets are static and no longer challenging, allowing current Large Language Models (LLMs) to perform well without truly revealing their capabilities and shortcomings. To address this problem, we propose a new data augmentation framework to automatically build more challenging NLI datasets based on existing datasets, by iteratively fusing rich facts into the premise and hypothesis of an NLI instance. We use a strict fact filter to ensure that fused facts are non-contradictory and non-redundant. Applied to SNLI and MNLI, our augmentation substantially increases data length and complexity, and the performance of a range of LLMs on the augmented datasets drops significantly (up to 30%). Ablation experiments and human quality checks confirm the high quality of the generated data.

## 1 Introduction

Natural Language Inference (NLI) is a foundational task in language understanding: given a premise, decide whether a hypothesis is entailed, contradicted, or neutral (Dagan et al., 2006). Despite its centrality, we argue that current benchmarks face two core weaknesses: **simplicity** and **staticity**, resulting in modern systems to often report very high scores on standard NLI benchmarks, failing to reveal genuine performance on NLI tasks and understanding capacity.

Regarding **simplicity**, widely used datasets such as the Stanford Natural Language Inference corpus (SNLI; Bowman et al., 2015) and the Multi-Genre Natural Language Inference corpus (MNLI; Williams et al., 2018) concentrate on short, lexically cued items that hinge on single, local relations. Such items underrepresent longer contexts, multi-fact dependencies, and diverse semantic phenomena, making it easy for models to succeed without robust content coverage or careful semantic comparison (Gururangan et al., 2018; Poliak et al., 2018; McCoy et al., 2019).

Regarding **staticity**, these datasets are typically collected once (via crowdsourcing/annotation) and then frozen. As a result, they do not keep pace with evolving knowledge, domains, or usage, and they offer no mechanism to systematically adjust difficulty or enrich instances over time—an issue evident across widely used static resources such as SICK (Marelli et al., 2014), and XNLI (Conneau et al., 2018).

To address the issue of *simplicity*, earlier studies turned to human-in-the-loop adversarial collection: by iteratively eliciting model failures, they constructed more challenging and diverse examples. ANLI (Nie et al., 2020a) proceeds in rounds of human–model interaction to expose systematic weaknesses; Dynabench (Kiela et al., 2021) extends this paradigm into a platform for continuously updating datasets against deployed models; and WANLI (Liu et al., 2022) leverages worker–model collaboration to generate candidates that are subsequently filtered by humans. While such approaches improve hard-example coverage, they remain static once collected, incur substantial annotation costs, and do not provide a mechanism for continuously injecting verifiable evidence or systematically scaling difficulty.

In parallel, model-based synthetic augmentation has rapidly developed: large language models are prompted to generate new premise–hypothesis pairs, sometimes with templates or instructions, followed by lightweight filtering. Hosseini et al. (2024) showed that large-scale synthetic NLI can enhance domain generalization; more broadly, recent surveys confirm the potential of LLM-driven

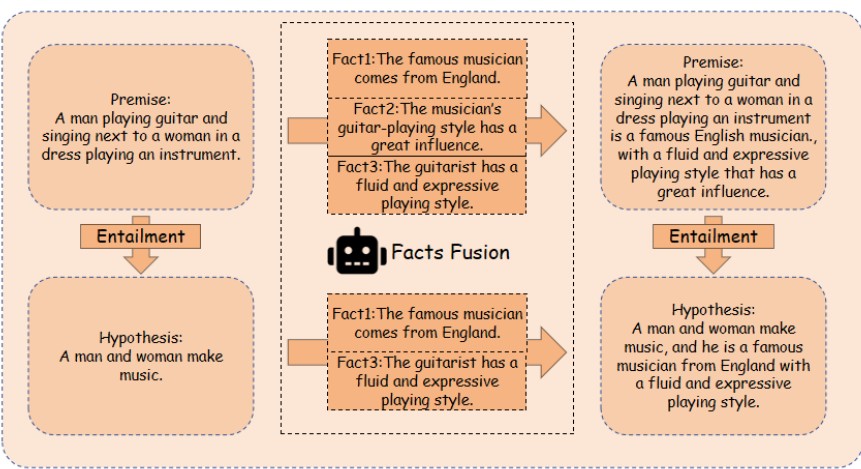

Figure 1: A one-level augmentation case with selected Facts

synthetic data to reduce annotation cost and expand coverage across domains and longer contexts (Nadas et al., 2025). However, such methods rely heavily on the generative model itself, often introducing unverifiable content, label drift, and stylistic artifacts. Even with partial human review, they lack explicit alignment with external evidence, making controllable difficulty and traceability difficult to guarantee.

To fully solve the problem,we propose a **dynamic, automated, fact-enhanced augmentation framework** that operates on existing NLI samples. For each (premise, hypothesis, label), we (i) retrieve premise-conditioned evidence from Wikipedia to ground the instance in verifiable content; (ii) filter candidate facts to ensure non-contradiction and non-redundancy so that labels are preserved; and (iii) fuse accepted facts into the premise and a conservative subset into the hypothesis. A multi-level framework enables tunable difficulty by progressively composing more evidence while maintaining label fidelity. Consequently, evidence-backed fusion directly addresses simplicity by increasing length and semantic richness with real, citable content, and mitigates staticity by enabling ongoing, retrieval-driven updates and depth control—without re-annotating from scratch. Figure 1 shows an example with one-level augmentation.

Applied to SNLI (Bowman et al., 2015) and MNLI (Williams et al., 2018), our augmentation (i) reveals consistent, depth-controlled performance drops across diverse model families (up to $30\%$ at higher depths) ; (ii) expands example length by multiple folds (up to $10\times$) and increases semantic density via multi-fact composition; and (iii) preserves labels and instance coherence, as verified by ablations and human evaluation—thereby exposing substantial headroom that static benchmarks conceal. To summarize, we list our main contributions as follows:

- We introduce a dynamic and automated fact-enhanced augmentation framework that enriches existing NLI items via a retrieval–filtering–fusion process, which preserves original labels, and enables depth-controlled difficulty.

- We apply our augmentation framework on SNLI and MNLI datasets and release new challenging benchmarks to benefit the research community.

- We conduct a comprehensive evaluation across a range of LLM-based NLI models, showing consistent, depth-controlled performance degradation that highlights a substantive challenge understated by current static benchmarks.

## 2 RELATED WORK

### 2.1 CLASSIC NLI BENCHMARKS AND HUMAN-IN-THE-LOOP CURATION

The Stanford Natural Language Inference corpus (SNLI; Bowman et al. (2015)) first established a large-scale, three-way classification benchmark, but its short, lexically cued pairs made it easy for models to exploit shallow cues rather than deep inference. The Multi-Genre NLI corpus (MNLI; Williams et al. (2018)) broadened coverage across genres, yet still preserved relatively short contexts, leaving similar vulnerabilities. The SICK dataset (Marelli et al., 2014) focused on compositional semantics, but its small size limited robustness. XNLI (Conneau et al., 2018) extended MNLI to 15 languages, enabling cross-lingual evaluation but again freezing into a static test set. HANS (McCoy et al., 2019) directly targeted heuristic shortcuts, showing that high accuracy on SNLI/MNLI does not imply robust inference. DocNLI (Yin et al., 2021) scaled inference to full documents, though without mechanisms for dynamic difficulty control.

To increase difficulty, ANLI (Nie et al., 2020a) introduced iterative adversarial collection where annotators probe system weaknesses, raising difficulty but at significant annotation cost. Dynabench (Kiela et al., 2021) generalized this into a platform for continuous interactive collection, though instances ultimately re-freeze into fixed test sets. ChaosNLI (Nie et al., 2020b) proposed retaining distributions over human judgments, highlighting interpretive variability, and Jiang & Pavlick (2022) further analyzed sources of label disagreement, but such work still lacks attached, verifiable evidence for each instance. In sum, while human-in-the-loop methods raise challenge and capture subjectivity, they remain expensive, hard to scale, and insufficient for long-context inference.

### 2.2 SYNTHETIC AUGMENTATION FOR NLI

Synthetic generation has emerged as a practical response to data scarcity. STraTA (Vu et al., 2021) integrates self-training with LM-generated premise–hypothesis pairs, increasing volume but offering no guarantees of evidence alignment. The GAL framework (He et al., 2022) likewise generates unlabeled text and relies on teacher models for annotation, lowering labeling cost while risking label drift. Li et al. (2023) provide a systematic analysis showing that synthetic data can improve classification in some settings, yet task subjectivity moderates gains and raises concerns about unverifiable artifacts. WANLI (Liu et al., 2022) combines automatic filtering with selective human review after GPT-3 expansion of "challenging pockets" in MNLI, achieving strong out-of-domain improvements but at renewed human cost. More recent work emphasizes domain-diverse augmentation (Hosseini et al., 2024), underscoring the promise of scaling with LLMs while exposing persistent issues of model priors and style biases.

## 3 METHOD

Natural Language Inference (NLI) determines the relationship between a premise and a hypothesis(Dagan et al., 2006). The outcome is one of three labels. *Entailment* means the premise provides sufficient information to conclude the hypothesis is true. *Contradiction* means the premise provides sufficient information to conclude the hypothesis is false. *Neutral* means the premise neither supports nor rules out the hypothesis.

### 3.1 OVERVIEW

Our goal is to automatically transform standard NLI samples into more challenging ones via adding verified facts to the premise and to the hypothesis in a multi-level manner. Given an input $(p, h, l)$ with premise $p$, hypothesis $h$, and label $l \in \{entailment, neutral, contradiction\}$, we iteratively expand $p$ and $h$ with external evidence while preserving logical consistency and the original label. After $L$ levels, we obtain an enriched sample $(p^{(L)}, h^{(L)}, l)$ in which $p^{(L)}$ and $h^{(L)}$ incorporate verified new facts.

As shown in Fig. 4, our method consists of three steps at every level: (i) Facts acquisition — Retrieve clean, verifiable facts for augmentation. (ii) Truth-set and graph filtering — Ensure no redundance and contradiction by removing conflicts and duplicates facts pairs. (iii) Premise and hypothesis

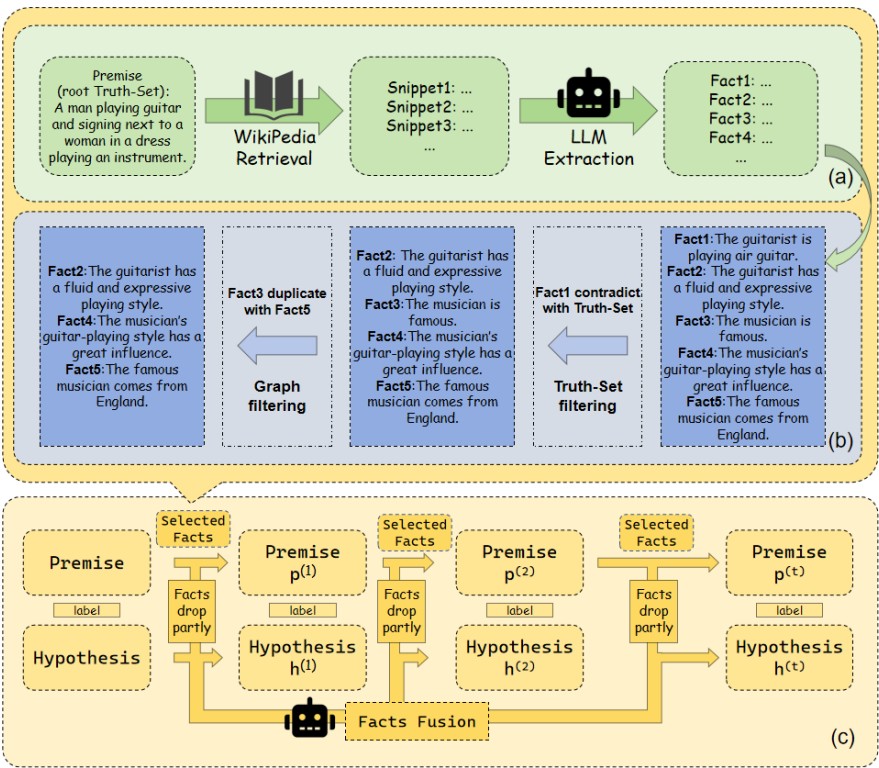

Figure 2: Overview of the full framework. (a) Premise-conditioned retrieval and fact extraction. (b) Truth-set and graph-based filtering of facts, where the initial truth set is derived from the original premise. (c) The complete multi-level augmentation framework; facts drop partly denotes the subset of facts retained into the hypothesis relative to the premise.

fusion — Merge selected facts into the premise and hypothesis while preserving the original label. We present details of each step as follows.

## 3.2 FACTS ACQUISITION

To select corpora closely related to premise for constructing the snippet pool used in later fusion , at level $t$, we use the former premise $p^{(t-1)}$ as the query to retrieve candidate Wikipedia pages. From each page, we take the introductory summary and split it into sentences. We compare sentence embeddings to the query embedding via cosine similarity. We keep sentences with score $\geq \tau$, where $\tau$ is a fixed, moderate threshold. The retained sentences form the snippet pool $\mathbb{P}^{(t)}$. This pool is the only evidence used in subsequent steps (Fig. 4a).

Given only $\mathbb{P}^{(t)}$, we apply a strong LLM (GPT-4o) to extract atomic facts. Specifically, we introduce a carefully designed prompt to ensure that the extracted facts contain no external knowledge or inference and are grounded in exactly one snippet from $\mathbb{P}^{(t)}$. The output is the candidate fact set $\mathbb{C}^{(t)}$. Full prompt templates with more rules appear in Appendix B.1. Further implementation details are provided in Section §4.1.

## 3.3 TRUTH-SET AND GRAPH FILTERING

To ensure the facts to be fused do not conflict with or duplicate the current premise, we maintain a truth set $\mathbb{T}^{(t)}$ including facts fused at earlier levels and extracted from the original premise. We then construct an entailment graph $\mathcal{G}$ to check pairwise relations among candidate facts and remove conflicting or redundant items.

At level $t$, the candidates set $\mathbb{C}^{(t)}$ is first checked against truth-set: for each $f \in \mathbb{C}^{(t)}$, a lightweight NLI model compares $f$ with every element of $\mathbb{T}^{(t-1)}$ and assigns a relation in {entailment, contradiction, neutral}. Any $f$ that contradicts an element of $\mathbb{T}^{(t-1)}$ is discarded. Any $f$ that is entailed by $\mathbb{T}^{(t-1)}$ is marked as redundant and discarded. Facts that are neutral with respect to all elements of $\mathbb{T}^{(t-1)}$ are retained. Denote the retained facts by $\mathbb{V}^{(t)}$.

We discard conflicts and redundancies among the retained facts by constructing an undirected graph $\mathcal{G}^{(t)}$ on $\mathbb{V}^{(t)}$. An edge connects $u, v \in \mathbb{V}^{(t)}$ if and only if their relation is *neutral*; pairs in *entailment* or *contradiction* are excluded to prevent conflict and redundancy. We take the **maximum clique** of the neutral-edge graph $\mathcal{G}^{(t)}$ (constructed on $\mathbb{V}^{(t)}$) as the level-$t$ fact set, denoted $\mathbb{F}^{(t)}$. A clique guarantees pairwise neutrality among all selected facts; choosing the largest such clique yields the broadest subset that is simultaneously non-conflicting and non-redundant under our criterion.

We then update the truth set by

$$\mathbb{T}^{(t)} \;=\; \mathbb{T}^{(t-1)} \cup \mathbb{F}^{(t)}.$$

This two-stage procedure removes conflicts with prior facts, suppresses items already implied by history, and at each level admits a maximal cluster of pairwise neutral (non-entailed, non-contradictory) facts, thereby preserving global consistency as $\mathbb{T}$ grows (Fig. 4b). Detailed algorithms are provided in Appendix C.

### 3.4 PREMISE AND HYPOTHESIS FUSION

At each level $t = 1, \ldots, d$, we have selected a set of novel facts $\mathbb{F}^{(t)}$ from the candidates after flitering. We then update the premise by fusing $\mathbb{F}^{(t)}$ with the previous anchor $p^{(t-1)}$ via a constrained LLM prompt to produce a short paragraph $p^{(t)}$. By construction, $p^{(t)}$ entails $p^{(t-1)}$, fully covers the content in $\mathbb{F}^{(t)}$, and introduces no information beyond $p^{(t-1)} \cup \mathbb{F}^{(t)}$.

In contrast, for the hypothesis, we always enhance from the original hypothesis $h$. At level $t$, for each $i \in \{1, \ldots, t\}$ we select a subset $\widehat{\mathbb{F}}^{(i)} \subseteq \mathbb{F}^{(i)}$ from level $i$, and then form the aggregate $\mathbb{F}^* = \bigcup_{i=1}^{t} \widehat{\mathbb{F}}^{(i)}$. We fuse $\mathbb{F}^*$ with $h$ to obtain $h^{(t)}$, which entails $h$, fully reflects $\mathbb{F}^*$, and adds nothing beyond $h \cup \mathbb{F}^*$. The detailed prompt templates are provided in Appendix B.2. The final enhanced sample at level $t$ is $(p^{(t)}, h^{(t)}, l)$.

Note that $h^{(t)}$ is always generated directly from the original hypothesis $h$ (rather than from $h^{(t-1)}$) to avoid semantic drift and label reinterpretation across levels. This also enables controlled difficulty scaling: by aggregating $\mathbb{F}^*$ at higher $t$, we inject more verified, decision-bearing content into the hypothesis without compounding generation artifacts.

Finally, across the $L$ levels we obtain $L$ enhanced premise–hypothesis pairs $\{(p^{(t)}, h^{(t)}, l)\}_{t=1}^{L}$ (Fig. 4(c)). We provide a complete case in Appendix G.

## 4 EXPERIMENTS

### 4.1 EXPERIMENTAL SETUP

**Augmentation Data Source.** We apply our framework to *all* instances from the following official splits (no sub-sampling): **SNLI** (Bowman et al., 2015) `test`; **MNLI** (Williams et al., 2018) `validation_matched` and `validation_mismatched`. Dataset sizes are shown in Table 1.

Table 1: Dataset sizes used in augmentation.

| Dataset | Split | Size |
|---------|-------|------|
| SNLI | test | 9,824 |
| MNLI | validation_matched | 9,815 |
| MNLI | validation_mismatched | 9,832 |

We choose these splits for two practical reasons: (i) many widely used pretrained or instruction-tuned models have been (directly or indirectly) fine-tuned on the training portions of SNLI/MNLI, so evaluating on the official held-out splits mitigates train–test contamination; and (ii) the public test split of MNLI does not release gold labels, hence evaluation customarily uses the labeled validation_matched and validation_mismatched splits instead.

Both corpora adopt a unified instance schema $(p, h, l)$ comprising a *premise*, a *hypothesis*, and a *label*. In their original construction protocols, human annotators wrote hypotheses conditioned on sampled premises and assigned labels to the resulting pairs.

**Implementation Details.** We deliberately separate *generation* from *filtering*. The only strong LLM in our framework is GPT-4O (OpenAI, 2023), used for two operations: (i) snippet-faithful fact extraction and (ii) constrained *fusion* to produce both the level-wise enriched premise and the final hypothesis.

All screening steps rely on lightweight models for scalability: sentence-level *relevance* is computed with ALL-MINILM-L6-V2 embeddings (Reimers & Gurevych, 2019; Wang et al., 2020) (with simple lexical hygiene), and *entailment/contradiction/neutral* edges in the truth-set graph are assigned by a compact cross-encoder based on DEBERTA-V3-BASE (He et al., 2021b;a), following standard cross-encoder reranking practice (Nogueira & Cho, 2019). This design emphasizes GPT-4o for content generation while delegating relevance and entailment filtering to efficient small models.

We intentionally isolate generation from inference to avoid leakage and bias. Premise updates are generated only from the previous anchor and selected facts, and hypotheses are generated only from the root hypothesis and the aggregated facts—never from any premise text. Gold labels remain fixed, and E/C/N decisions (and relevance scoring) are handled by lightweight discriminative models rather than GPT-4o. This separation reduces label and cross-side leakage, keeps generation orthogonal to the core reasoning task, and yields more impartial examples.

**Tested Models.** We evaluate seven models—ROBERTA-LARGE (Liu et al., 2019), DEBERTA-V3-LARGE (He et al., 2021a), GPT-4O (OpenAI, 2023), LLAMA-3-8B-INSTRUCT (Grattafiori et al., 2024), QWEN2.5-14B-INSTRUCT (Yang et al., 2024), DEEPSEEK-V3 (Liu et al., 2024), and DEEPSEEK-R1 (Guo et al., 2025)—on the original datasets ($L=0$, *orig*) on our dynamically enhanced versions at three *levels* ($L=1, 2, 3$).

## 4.2 MAIN RESULTS

Across all datasets and models, augmentation produces a *consistent accuracy drop* that grows with the *level*, as shown in Table 2, indicating that the enhanced sets increase task hardness while preserving labels.

Performance decays approximately monotonically from $L=0 \rightarrow 1 \rightarrow 2 \rightarrow 3$, with the largest stepwise decline typically at the first enhancement ($0 \rightarrow 1$) and smaller but still nontrivial declines from $1 \rightarrow 2$ and $2 \rightarrow 3$.

Across both SNLI and MNLI, accuracies drop sharply at the first enhancement step ($L=1$) and then decline more gradually at higher levels (Table 2); GPT-4O follows this pattern, as do other models. This indicates that a single shallow enhancement ($L:0 \rightarrow 1$) already induces most of the added difficulty, while deeper levels mainly accumulate incremental complexity. Taken together, the results show that our framework strengthens understanding and inference from the outset, with deeper levels providing refinement and further improvements.

To summarize, dynamic fact-enhanced augmentation consistently reduces accuracy across models and datasets in a level-controlled manner, turning standard NLI benchmarks into stronger stress tests for text inference abilities.

## 4.3 ANALYSIS OF DIFFERENT LEVELS

We treat the original data as level $L=0$ (*baseline*) and track length as enhancement level rises to $L=1, 2, 3$. We report *word* counts for both premise and hypothesis .

Across datasets, baselines are short: premises average 14–20 words and hypotheses 7–10. By $L=3$, premises reach 101–118 and hypotheses 70–76. In relative terms, premise length grows by 5–8× and hypothesis length by 7–10×, with steady gains at each level (Fig. 3a), indicating increased semantic load on both sides while preserving gold labels.

Table 2: Accuracy on Enhanced Datasets at Varying Levels ($L$)

| Model | SNLI | | | | MNLI | | | |
|---|---|---|---|---|---|---|---|---|
| | $Acc_{L=0}$ | $Acc_{L=1}$ | $Acc_{L=2}$ | $Acc_{L=3}$ | $Acc_{L=0}$ | $Acc_{L=1}$ | $Acc_{L=2}$ | $Acc_{L=3}$ |
| RoBERTa-large | 89.3 | 70.8 | 64.7 | 60.8 | 90.4 | 73.7 | 70.4 | 68.1 |
| DeBERTa-v3-large | 92.4 | 77.1 | 69.0 | 67.2 | 90.1 | 74.2 | 71.9 | 69.7 |
| Llama-3-8B-Instruct | 55.6 | 43.2 | 41.6 | 41.1 | 64.3 | 55.3 | 53.4 | 52.3 |
| Qwen2.5-14B-Instruct | 82.4 | 64.7 | 59.4 | 56.7 | 82.3 | 62.3 | 59.5 | 58.8 |
| GPT-4o | 84.8 | 62.1 | 58.7 | 55.3 | 83.5 | 67.5 | 67.0 | 66.7 |
| DeepSeek-V3 | 81.3 | 64,8 | 62.7 | 61.5 | 82.3 | 69.7 | 68.5 | 67.0 |
| DeepSeek-R1 | 78.9 | 60.2 | 56.6 | 53.1 | 82.2 | 71.7 | 70.9 | 63.3 |
| *Average* | 80.7 | 63.3 | 59.0 | 56.7 | 82.2 | 67.8 | 65.9 | 63.7 |

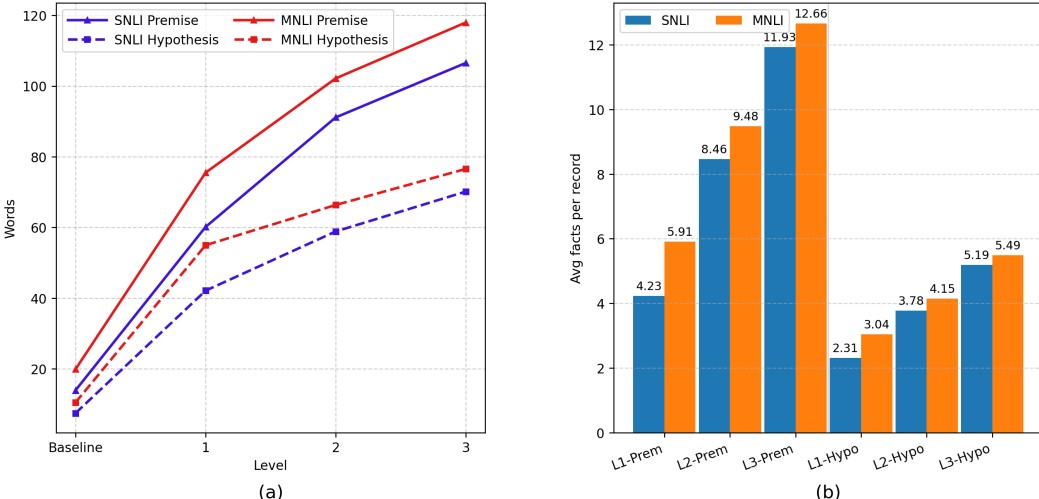

(a)

(b)

Figure 3: **Length growth and fact fusion by level.** **(a)** Average *word* counts vs. level ($L=0$ baseline, then $L=1, 2, 3$) for SNLI and MNLI, shown separately for *premise* (solid) and *hypothesis* (dashed). **(b)** Average fused facts per record at levels $L=1, 2, 3$ on both sides (Prem/Hypo); numbers above bars indicate the exact averages. Baseline ($L=0$) is omitted for clarity.

Let $\Delta_{\text{words}}^{(L-1 \to L)} = \overline{\text{words}}_L - \overline{\text{words}}_{L-1}$. The increment peaks at the first step ($L:0 \to 1$) and then tapers off at higher levels, for both premises and hypotheses.

In the fact-tracked subset, both premise-side and hypothesis-side fact counts rise monotonically with depth. Crucially, deeper levels do not merely pad the premise; they consolidate verified facts into the hypothesis, increasing the proportion of decision-bearing content and thereby strengthening semantic pressure (Fig. 3b).

From both word counts and the number of facts, we observe the largest gain from $L=0 \to 1$, validating the method's design; subsequent levels still deliver improvements, albeit diminishing. Crucially, this growth in factual load is mirrored by consistent declines in model accuracy across models and metrics—the steeper drop at $L=1$ followed by continued reductions at higher levels. The alignment between rising fact counts and falling accuracy indicates that fact-based augmentation is an effective mechanism for increasing task difficulty, and that the level parameter $L$ provides reliable, predictable control over that difficulty.

## 4.4 ABLATIONS

**Setup.** To isolate the effect of the *truth set and entailment-graph filter*, we re-run the framework with the filter disabled (NO-FILTER)—which means we no longer maintain a truth set or an entailment graph—while holding everything else fixed. We report results at $L=3$ (primary endpoint) and include level sensitivity at $L=1, 2$.

**Results.** The truth set and entailment-graph filter enforces global consistency by vetoing contradictions and suppressing redundancies. When we disable it, three failure modes emerge: (i) *label drift*—often driven by two mechanisms: items originally labeled *Contradiction* flip to *Neutral* or *Entailment* when we inject premise-entailed facts into both the premise and hypothesis (neutralizing the contradiction), and items originally labeled *Entailment* flip to *Contradiction* when conflicting facts are introduced; (ii) *corpus conflicts*—predominantly mismatches in time, location, or participants/roles (e.g., inconsistent dates, places, or entities across fused sources), as well as cross-source negations; and (iii) *redundant fusion*—already-entailed content is restated via near-duplicates or mechanical repetition across segments, inflating length (and sometimes superficial coverage) without adding genuine new information.

Consequently, we report NO-FILTER to exhibit a larger accuracy drop as shown in Table 3, to appear superficially "harder"—*together with* degraded coherence: reduced label preservation, increased internal contradiction, and elevated redundancy.

Table 3: Model accuracies at $L=0$ and $L=3$ under the Full framework and the NO-FILTER ablation.

| Model | $\text{Acc}_{L=0}(\%)$ | $\text{Acc}_{L=3, \text{Full}}(\%)$ | $\text{Acc}_{L=3, \text{No-Filter}}(\%)$ |
|---|---|---|---|
| RoBERTa-large | 90.3 | 59.3 | 26.0 |
| GPT-4o | 83.3 | 56.0 | 31.7 |

In short, removing the filter makes instances look harder primarily by introducing inconsistency and noise rather than principled difficulty. To quantitatively assess quality more accurately, we redefine our evaluation metrics; details are provided in Section §4.5.

## 4.5 EX-POST HUMAN AUDIT

**Design.** Following common practice in evaluation studies for NLI, fact verification, and factuality assessment, we run a *blinded, stratified, double-annotation* audit (Bowman et al., 2015; Williams et al., 2018; Thorne et al., 2018; Nie et al., 2020a; Kiela et al., 2021; Maynez et al., 2020; Pagnoni et al., 2021). We sample enhanced items across levels $L \in \{1, 2, 3\}$ with proportional stratification by dataset (SNLI/MNLI), fixing the number of items per stratum. Annotators see only the root premise/hypothesis/label, the enhanced texts, and the exact evidence snippets used during enhancement; they are blind to model predictions and to whether an item comes from the full framework or an ablation. Each item is independently labeled by two annotators; disagreements are adjudicated by a senior third annotator. We include attention checks and a brief calibration round with gold items before the main task (Nangia et al., 2021). We report inter-annotator agreement using Cohen's $\kappa$ for categorical questions and linearly weighted $\kappa$ for ordinal scales. Unless noted otherwise, percentages are macro-averaged across strata.

**Annotation Task and Metrics.** Beyond accuracy on enhanced sets, we report five automatic diagnostics—*label preservation* (agreement with the baseline label), *internal contradiction* (fraction of enhanced items with conflicts), *redundancy* (fraction of facts entailed by others), *factuality* (fraction of retained facts entailed by the final text), and *readability* (Likert)—and treat them as automatic proxies of quality. Complementarily, a stratified human audit elicits a single five-part judgment per item, aligned with these dimensions: (i) *label preservation* (E/N/C)—whether the enhanced pair maintains the original label; (ii) *internal contradiction / corpus conflict* (yes/no)—whether any selected facts contradict one another or the anchor premise, judged as evidence-based contradiction in the sense of fact verification Thorne et al. 2018; (iii) *redundant fusion* (none/some/many)—the extent to which the enhanced text restates or paraphrases already-entailed content (cf. factuality audits Maynez et al. 2020; Pagnoni et al. 2021); (iv) *factuality w.r.t. evidence* (supported/partially/unsupported)—whether claims in the enhanced text are supported by the provided facts (as in FEVER Thorne et al. 2018); and (v) *readability* (Likert 1–5)—clarity and grammaticality of the enhanced text. This joint design lets human judgments be directly comparable, clarifying whether accuracy drops reflect principled, multi-level difficulty or artifacts of incoherent editing.

The complete annotation guidelines are provided in Appendix H.1.

**Sampling.** We pre-specify a per-stratum target of 240 items to detect data quality. More sampling details are provided in Appendix H.2

Table 4: Ex-post human audit by level ($L \in \{1, 2, 3\}$), macro-averaged over datasets.

| Level | Label pres. (%) | Conflict (%) | Redundancy (%) | Factuality (%) | Readability |
|-------|-----------------|--------------|----------------|----------------|-------------|
| L=1 | 98.3 | 1.7 | 11.7 | 100.0 | 4.1 |
| L=2 | 96.7 | 5.0 | 10.0 | 100.0 | 3.9 |
| L=3 | 95.0 | 6.7 | 13.3 | 98.3 | 3.9 |

Table 5: Full framework vs. NO-FILTER at $L=3$ , macro-averaged over datasets.

| Variant | Label pres. (%) | Conflict (%) | Redundancy (%) | Factuality (%) | Readability |
|---------|-----------------|--------------|----------------|----------------|-------------|
| **Full** | 95.0 | 6.7 | 13.3 | 98.3 | 3.9 |
| **No-Filter** | co | 31.7 | 28.3 | 98.3 | 3.3 |

**Results and Analysis.** Table 4 summarizes level-wise outcomes for the full framework. The audit supports that our augmentation yields hard yet high-quality data. Label preservation remains high across levels; internal conflicts are rare under the full framework; and redundancy is controlled. According to the annotators' reports, the repetition mainly stems from describing the same concept in different ways rather than from mechanical duplication, which is acceptable to some extent. Readability decreases slightly with level

Table 6: Inter-annotator agreement (IAA). Cohen's $\kappa$ for categorical tasks; linearly weighted $\kappa$ for ordinal scales.

| Dimension | $\kappa$ |
|-----------|----------|
| Absolute NLI label (E/N/C) | 0.94 |
| Internal contradiction (yes/no) | 0.83 |
| Redundancy (none/some/many) | 0.68 |
| Factuality vs. fact list (3-way) | 0.98 |
| Readability (Likert 1–5; weighted) | 0.57 |

but stays in the "clear" range. These trends align with automatic diagnostics and the accuracy drops reported in §4.2 .

Table 5 shows that, at $L=3$, disabling the filter (NO-FILTER) leads to a clear degradation in corpus quality . The NO-FILTER induces pronounced label drift, frequent contradictions, and elevated repetition alongside poorer readability. Factuality remains comparable, substantiating the effectiveness of our fusion mechanism. In short, the quality drop under NO-FILTER is unacceptable for evaluation or downstream use, and we therefore treat the filtered framework as the only reliable setting.

Final IAAs are reported in Table 6. Overall agreement is consistently high, demonstrating stable and replicable judgments across dimensions.

In summary, human evaluation shows that our augmentation raises difficulty while preserving high data quality. The filter design is crucial to this outcome: with the filter enabled, coherence is maintained (labels are preserved, contradictions are rare, and redundancy is controlled), whereas the NO-FILTER ablation exhibits a clear quality drop.

## 5 CONCLUSION

We presented a *dynamic, automated fact-based* augmentation framework that converts existing NLI examples into new challenging ones. By iteratively retrieving, distilling, filtering, and fusing atomic facts, our method increases semantic richness while preserving original labels. A tunable level $L$ offers a simple knob to scale difficulty: even shallow enhancement suppresses shortcut signals, whereas deeper enhancement compounds inference demands. Across SNLI and MNLI, the augmented data produce consistent, monotonic accuracy reductions, suggesting improved discriminative power for inference ability. Ablations confirm that the truth-set and graph filter is essential for quality, and a blinded human audit supports label preservation, low internal conflict, and strong evidence grounding.

This shift from one-shot curation to iterative, retrieval-driven enrichment offers a practical path to benchmarks that better measure inference and evolve with models. Future work may expand beyond Wikipedia to domain-specific and multilingual corpora, and explore learned fusion together with causal and adversarial probes to test understanding and inference capabilities more directly.

STATEMENT

**Ethics Statement.** We adhere to the ICLR Code of Ethics. Our work augments public NLI benchmarks (SNLI, MNLI) with automatically generated fact-enhanced variants and conducts a small-scope human evaluation. All annotators were adults who provided informed consent and could withdraw at any time; no personally identifiable information was collected. Source texts come from publicly available corpora under their original licenses.

**Reproducibility Statement.** We submit a compressed archive containing the code and datasets required to reproduce all results end-to-end. The prompts embedded in our code are identical to those documented in Appendix B; the filtering algorithm follows Appendix C; and the dataset specification and release procedures follow Appendix G.

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

## A  LLM USAGE

Large language models were used solely for grammar correction and stylistic polishing of the manuscript text.

## B  PROMPT

### B.1  FACT EXTRACTION

```
[Task Description]
You are a meticulous information extractor. Your sole job is to read the
given SNIPPETS and return only literal, atomic facts present in those
snippets.You must not infer, generalize, add outside knowledge.

[Inputs]
{SNIPPETS}

[Rules]
1) Source-only: use only snippet content; preserve tense, modality,
quantifiers, and negation exactly as written.
2) Minimality: each fact must be irreducible (cannot be further split).
3) No cross extraction: each fact must be fully supported by a single
snippet. Do not combine evidence across multiple snippets.
4) No ambiguous references: avoid vague or deictic pronouns (e.g., it,
they, he, she, this, that, former, latter, here, there) ; replace with
the explicit noun phrase from the snippet.
5) If nothing is extractable, return {"facts": []} exactly.

[Output JSON]
{
  "facts": [
    {
      "text": "<atomic fact1>",
    },
    {
      "text": "<atomic fact2>",
    }
    ...
  ]
}
```

## B.2 PREMISE AND HYPOTHESIS FUSION

The fusion procedure for the premise and the hypothesis is identical.In both cases, the to be fused p and h are provided as ANCHOR to the fusion step.

```
[Task Description]
You are a precise composer. Begin with the given ANCHOR text verbatim,
then add ONLY the supplied atomic facts from FACT_LIST_JSON. Do not
introduce any new information beyond those facts.

[Inputs]
{ANCHOR_TEXT}
{FACT_LIST_JSON}

[Rules]
1) Anchor-centric: write a short paragraph that revolves around
ANCHOR_TEXT.
2) Facts-only: integrate all and only the provided facts; each fact
appears once.
3) Fidelity: preserve polarity, modality, time expressions, and named
entities as given.
4) Minimal glue: use neutral connectors ("Additionally,", "At <time>,", "
In <location>,", "Meanwhile,") without adding new information.

[Output JSON]
{
  "fusion_result": "<fused_paragraph>"
}
```

## C ALGORITHM

**Input:** Truth set $T$, candidate facts $F$, NLI model $\mathcal{M}$, thresholds $\tau_e, \tau_c$
**Output:** Maximum neutral clique $S^\star \subseteq F$ and directed entailment subgraph $G^\Rightarrow[S^\star]$

1: **Predicates:**
2: ENTAILS$(a, b) \triangleq \Pr_{\mathcal{M}}(a \Rightarrow b) \geq \tau_e$
3: CONTRADICTS$(a, b) \triangleq \Pr_{\mathcal{M}}(a \perp b) \geq \tau_c$
4: NEUTRAL$(a, b) \triangleq \neg$ENTAILS$(a, b) \wedge \neg$CONTRADICTS$(a, b)$

5: **Stage 1: Truth-set filtering**
6: $R \leftarrow \emptyset$
7: **for** $f \in F$ **do**
8:     **if** $\forall t \in T : \neg$CONTRADICTS$(t, f) \wedge \neg$ENTAILS$(t, f)$ **then**
9:         $R \leftarrow R \cup \{f\}$

10: **Stage 2: Entailment edges and neutral graph on** $R$
11: $E^\Rightarrow \leftarrow \emptyset; \quad Adj(u) \leftarrow \emptyset, \forall u \in R$
12: **for** $i = 1$ **to** $|R|$ **do**
13:     **for** $j = i+1$ **to** $|R|$ **do**
14:         $u \leftarrow R[i], \ v \leftarrow R[j]$
15:         **if** ENTAILS$(u, v)$ **then**
16:             $E^\Rightarrow \leftarrow E^\Rightarrow \cup \{(u \to v)\}$
17:         **if** ENTAILS$(v, u)$ **then**
18:             $E^\Rightarrow \leftarrow E^\Rightarrow \cup \{(v \to u)\}$
19:         **if** NEUTRAL$(u, v)$ **then**
20:             $Adj(u) \leftarrow Adj(u) \cup \{v\}; Adj(v) \leftarrow Adj(v) \cup \{u\}$

21: **Stage 3: Maximum neutral clique (Bron–Kerbosch + pivot)**
22: $S^\star \leftarrow \emptyset$
23: **procedure** BK_PIVOT$(C, P, X)$
24:     **if** $P = \emptyset$ **and** $X = \emptyset$ **then**
25:         **if** $|C| > |S^\star|$ **then**
26:             $S^\star \leftarrow C$
27:         **return**
28:     choose $u \in P \cup X$ maximizing $|P \cap Adj(u)|$
29:     **for each** $v \in P \setminus Adj(u)$ **do**
30:         BK_PIVOT$(C \cup \{v\}, P \cap Adj(v), X \cap Adj(v))$
31:         $P \leftarrow P \setminus \{v\}; \quad X \leftarrow X \cup \{v\}$
32: BK_PIVOT$(\emptyset, R, \emptyset)$

33: **Stage 4: Induced neutral subgraph**
34: $G^\Rightarrow \leftarrow$ subgraph of $(R, E^\Rightarrow)$ induced by $S^\star$
35: **return** $S^\star, G^\Rightarrow$

# D    ANALYSIS OF DIFFICULTY

The previous analysis shows that our multi-level augmentation substantially increases both sequence length and the number of fused facts, and that model accuracy consistently decreases with higher levels. A natural question is whether this difficulty is merely a side effect of longer, noisier inputs, or whether it stems from models having to infer with additional, semantically aligned facts. We address this with two controlled ablations that keep length and surface form comparable while manipulating the informational content of the inputs.

Table 7: Difficulty ablations: accuracy on the original data, FactNLI ($L$=1), a rewrite-only variant, and an unrelated-facts variant, all controlled to have comparable input lengths.

| Model | Original | FactNLI $L$=1 | Rewrite | Unrelated-Facts |
|---|---|---|---|---|
| GPT-4o | 84.8 | 62.1 | 78.6 | 80.7 |
| DeepSeek-V3 | 81.3 | 64.8 | 76.9 | 71.9 |
| DeBERTa-v3-large | 92.4 | 77.1 | 80.4 | 85.2 |
| Qwen2.5-14B-Instruct | 82.4 | 64.7 | 78.0 | 72.1 |

**Length-controlled rewrite-only ablation.**    First, we isolate the effect of sequence length. For each original $(p, h)$ pair, we construct a rewrite-only variant $(\tilde{p}, \tilde{h})$ by prompting a LLM to expand $p$ and $h$ into longer paraphrases that preserve their meaning, while targeting the same length distribution as our fact-augmented data (we use the $L$=1 setting as the target length, since matching $L$=3 would require extreme expansion and makes it difficult for the rewrite-only variant to preserve the original meaning without introducing new information). Crucially, no external facts are added in this condition: all content in $(\tilde{p}, \tilde{h})$ is a rephrasing or elaboration of the original sentence pair. We then evaluate the same models on the original data, the FactNLI $L$=1 data, and the rewrite-only (length-matched) data.

Across all evaluated setups (Table 7), rewriting to match FactNLI length leads to only modest drops relative to the original benchmark, whereas FactNLI-$L$=1 produces substantially larger declines for the same models and datasets. This pattern indicates that longer sequences and more tokens, by themselves, do not fully account for the observed difficulty; the specific way in which we inject external factual content matters.

**Unrelated-facts ablation.**    To test whether the injected facts themselves truly participate in inference and are responsible for the additional difficulty, rather than generic noise from extra sentences, we design an *unrelated-facts* control. Starting from the same original $(p, h)$ pairs, we follow the FactNLI fusion protocol but replace filtered Wikipedia facts with commonsense or encyclopedic statements that are deliberately unrelated to the premise and hypothesis (different entities, topics, or events), while keeping length and discourse style comparable to FactNLI. In other words, this variant adds irrelevant information around $(p, h)$.

As shown in Table 7, augmenting with unrelated factual sentences again yields only small changes in accuracy compared to the original data, while FactNLI causes a much larger drop on the same models. Since sequence length and the amount of added text are comparable across FactNLI and the unrelated-facts control, this result suggests that the extra difficulty is not driven simply by more context or more information, but by the presence of semantically aligned facts that interact with the entities and events in $(p, h)$ and must be selectively integrated or ignored.

Taken together, these two ablations show that FactNLI increases difficulty in a way that goes beyond one-dimensional length scaling. Models do not fail merely because the inputs are longer or noisier; they fail when they must jointly reason over the original premise–hypothesis pair and a large set of newly injected, entity-aligned facts, treating *all* atomic statements in the combined context as candidates for inference. This is precisely the kind of evidence that our augmentation is designed to introduce, and it explains why performance on FactNLI is substantially lower than on both the original benchmarks and length-matched controls.

# E    CONFUSION MATRICES

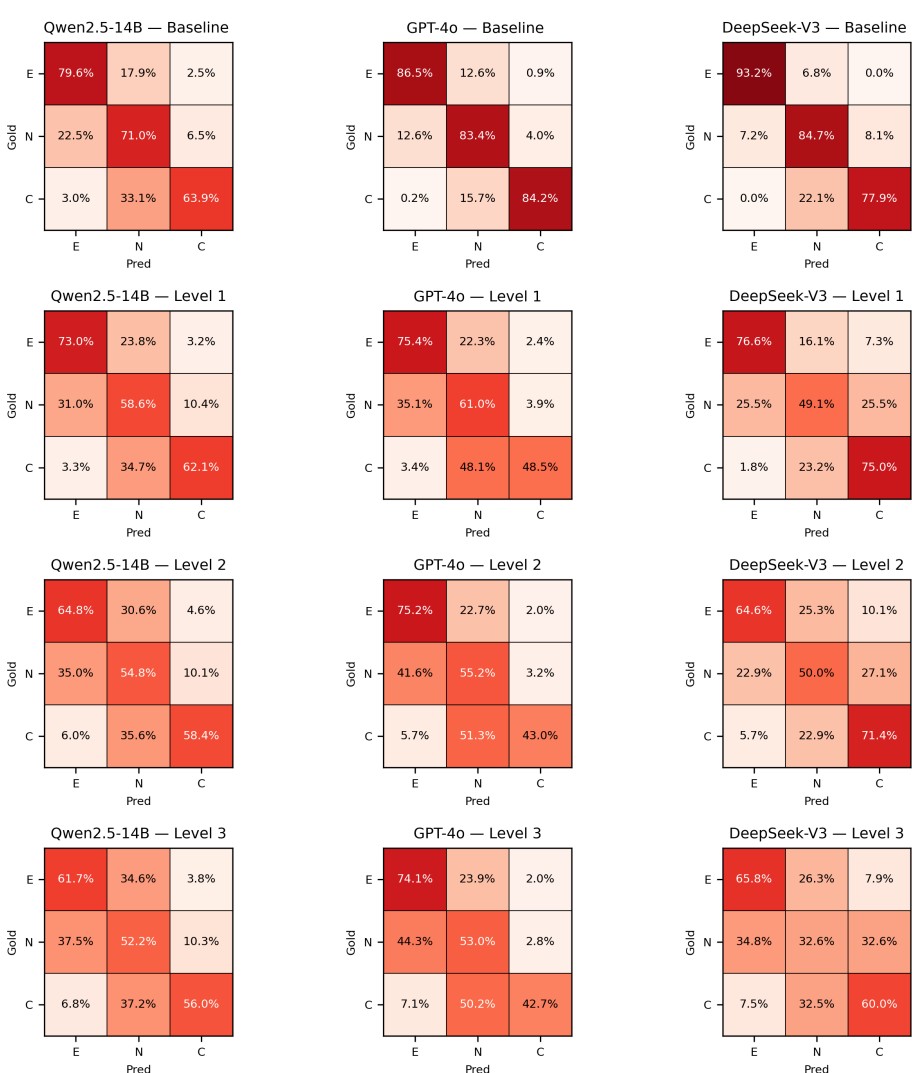

Figure 4: Confusion Matrices for Model Performance Across Different Levels and Models

# F    OPEN-SOURCE GENERATION

Our main experiments use GPT-4o as the backbone generator in the retrieve–filter–fuse framework. To verify that the framework is not tied to a single proprietary model, we additionally instantiate the generative components with the open-source **Qwen2.5-14B-Instruct** model on the SNLI dataset.

**Generation setup.**    We keep the retrieval procedures, fact-filtering steps, and label-preservation checks identical to the GPT-4o setup, and only replace the backbone generator with Qwen2.5-14B-Instruct . For each base SNLI pair and enhancement level $L=0, 1, 2, 3$, Qwen expands the original premise and hypothesis into longer, more discursive versions and integrates retrieved Wikipedia facts at levels $L=1, 2, 3$, ensuring entity and topic alignment. Prompts follow the same structure as in App. B, adapted to Qwen's chat template.

**Evaluation on Dataset Quality.**    We treat Qwen2.5-14B-Instruct as the NLI solver and evaluate it on the Qwen-generated SNLI variants , reporting accuracy drops in a manner consistent with the main experiments . The overall trend mirrors our findings with GPT-4o: performance degrades monotonically as the enhancement level increases, indicating that a open-source model also make SNLI substantially harder .

Table 8: Accuracy on Qwen-generated Datasets at Varying Levels ($L$)

| Model | $\text{Acc}_{L=0}$ | $\text{Acc}_{L=1}$ | $\text{Acc}_{L=2}$ | $\text{Acc}_{L=3}$ |
|---|---|---|---|---|
| GPT-4o | 84.8 | 55.3 | 49.7 | 50.0 |
| DeepSeek-V3 | 81.3 | 53.5 | 51.3 | 45.8 |
| Qwen2.5-14B-Instruct | 82.4 | 58.5 | 53.4 | 51.6 |

In addition, we conduct a small-scale manual audit like Sec 4.5. As Tabel 9 shown, open-source models show a slight decrease in label preservation, factual coverage, and readability compared to the proprietary model, they still maintain a high level of performance. The injected sentences are consistently topically aligned and logically compatible with the original premise and hypothesis. These findings demonstrate that our retrieve-filter-fuse framework is effective across different models, ensuring high-quality augmentation with preserved logical consistency and topic relevance.

Table 9: Ex-post human audit on two versions of datasets.

| Model | Label pres. (%) | Conflict (%) | Redundancy (%) | Factuality (%) | Readability |
|---|---|---|---|---|---|
| **GPT** | 96.7 | 4.5 | 11.7 | 99.4 | 4.0 |
| **Qwen** | 93.3 | 5.0 | 5.3 | 91.0 | 3.5 |

# G CASE STUDY

Table 10 shows the 3-level augmentation process of an example from SNLI with label *entailment*.

Table 10: Per-Level Fusion for an NLI Example: Facts, Premises, and Hypotheses (L0 → L3)

| Label | Text |
|---|---|
| Original Premise | A woman wearing a yellow tank top and white pants looking into a window on a city street. |
| Original Hypothesis | A woman wearing a tank top and pants looks through some glass. |
| L1 Facts | • Carnaby Street is a shopping street in Soho, City of Westminster.
• Carnaby Street is pedestrianised.
• Street fashion is associated with youth culture.
• Street fashion is seen in major urban centers. |
| L1 Premise | A woman wearing a yellow tank top and white pants is looking into a window on a city street, where street fashion, often associated with youth culture, is prominently displayed in major urban centers like Carnaby Street, a pedestrianised shopping street in Soho, City of Westminster. |
| L1 Hypothesis | A woman wearing a tank top and pants looks through some glass, embodying the street fashion that is often seen in major urban centers and is closely associated with youth culture. |
| L2 Facts | • Fashions from small urban youth groups influenced elite designers and mass-market manufacturers in the middle of the decade.
• Fashion examples include miniskirts, culottes, and go-go boots, along with less common PVC dresses.
• The 1960s fashion featured diverse trends and broke many traditions, adopting new cultures.
• London is situated on the River Thames in southeast England and has been a major settlement for nearly 2,000 years.
• London is the capital and largest city of England and the UK, with a population of 9,841,000 in 2025. |

| Label | Text |
|-------|------|
| L2 Premise | A woman wearing a yellow tank top and white pants is looking into a window on a city street, where street fashion, often associated with youth culture, is prominently displayed in major urban centers like Carnaby Street, a pedestrianised shopping street in Soho, City of Westminster. The 1960s fashion featured diverse trends such as miniskirts, culottes, go-go boots, and less common PVC dresses, breaking many traditions and adopting new cultures, influenced by small urban youth groups that impacted elite designers and mass-market manufacturers in the middle of the decade. London, situated on the River Thames in southeast England, has been a major settlement for nearly 2,000 years and had a population of 9,841,000 in 2025. |
| L2 Hypothesis | A woman wearing a tank top and pants looks through some glass, reflecting the influence of street fashion seen in major urban centers like London, which is situated on the River Thames in southeast England and has been a major settlement for nearly 2,000 years. Fashion examples include miniskirts, culottes, and go-go boots, along with less common PVC dresses, where fashions from small urban youth groups influenced elite designers and mass-market manufacturers in the middle of the decade. |
| L3 Facts | • Street fashion is linked to youth culture and is commonly found in major urban centers.
• The neighborhood is historically known for clothing production and has a high concentration of fashion-related uses.
• Magazines and newspapers often feature candid photos of individuals in urban, stylish clothing.
• Fashion capitals are cities with significant influence on the international fashion scene. |
| L3 Premise | A woman wearing a yellow tank top and white pants is looking into a window on a city street, where street fashion, often associated with youth culture, is prominently displayed in major urban centers like Carnaby Street, a pedestrianised shopping street in Soho, City of Westminster. Street fashion is linked to youth culture and is commonly found in major urban centers, and the neighborhood is historically known for clothing production with a high concentration of fashion-related uses. London, the capital and largest city of England and the UK, is located on the River Thames in southeast England, has been a major settlement for nearly 2,000 years, and had a population of 9,841,000 in 2025. Fashion capitals are cities with significant influence on the international fashion scene, and magazines and newspapers often feature candid photos of individuals in urban, stylish clothing. |

| Label | Text |
|-------|------|
| L3 Hypothesis | A woman wearing a tank top and pants looks through some glass, in a neighborhood historically known for clothing production, where street fashion is prevalent and magazines often feature candid photos of individuals in urban, stylish clothing, reflecting the influence of fashion capitals like London, which is situated on the River Thames in southeast England and has been a major settlement for nearly 2,000 years. Fashion examples include miniskirts, culottes, and go-go boots, along with less common PVC dresses. |

# H HUMAN AUDIT

## H.1 ANNOTATOR GUIDELINE

**Purpose.** Evaluate the *enhanced* NLI items for *absolute* quality. Judge: (i) the NLI relation, (ii) factual correctness *with respect to the provided fact list*, (iii) internal consistency, (iv) redundancy, and (v) readability. **Do not** use outside sources or personal knowledge.

**Materials per item.**

- **Sample texts:** a *premise* and a *hypothesis*.
- **Fact list:** 3–15 *atomic facts* (short, self-contained statements) used to construct or justify the sample.

**Core labeling principles (from standard NLI practice).**

1. **Judge the hypothesis relative to the premise**, not real-world truth.
2. **Use only minimal, text-licensed inference:** paraphrase, synonymy, simple hypernymy/hyponymy, obvious arithmetic, and straightforward temporal/order reasoning.
3. **Use the fact list as the only external support.** If a central claim is not supported or clearly entailed by the facts, mark it *unsupported*.
4. **Prefer certainty over plausibility.** If the premise does not guarantee truth or falsity, choose **Neutral**.

**How to read.**

1. Read the premise and hypothesis carefully.
2. Read the fact list: skim once for coverage, then re-read to verify specific claims in the texts against the facts.
3. **Factuality rule:** judge factuality *only* against the fact list. New content not supported (or clearly entailed) by the listed facts is *unsupported*.

**What to answer (absolute, five-part judgment).**

1. *Absolute NLI label (E/N/C):* choose **Entailment** (premise makes the hypothesis certainly true), **Contradiction** (certainly false), or **Neutral** (neither guaranteed nor refuted).
2. *Internal contradiction (yes/no):* mark "yes" if the premise and hypothesis (or content inside them) contradict themselves or each other, independent of the fact list.
3. *Redundancy (none/some/many):* rate repetition or trivial paraphrase without a new reasoning step: **None** (no noticeable repetition), **Some** (occasional), **Many** (frequent, length-inflating).
4. *Factuality w.r.t. fact list (supported/partially/unsupported):* check each *central claim* against the facts only.
5. *Readability (Likert 1–5):* 1 (unreadable), 2 (poor), 3 (adequate), 4 (good), 5 (clear and natural).

**Allowed vs. not allowed.**

- **Allowed:** paraphrase, synonymy, simple hypernymy/hyponymy, basic arithmetic, direct temporal reasoning, transitivity—when licensed by the text/facts.
- **Not allowed:** web search, outside knowledge, multi-hop world knowledge not in the facts, speculative assumptions.

**Submission checklist (per item).**

- Absolute NLI label (E/N/C).
- Internal contradiction: yes/no (+ short note if "yes").
- Redundancy: none/some/many (+ brief reason if "many").
- Factuality vs. fact list: supported/partially/unsupported (cite fact IDs if helpful).
- Readability: 1–5.

REFERENCE EXAMPLES

**A. NLI label (E/N/C) examples**

- **Entailment (E).**
  *Premise:* "The match was postponed due to heavy rain."
  *Hypothesis:* "Weather caused the match to be delayed."
- **Contradiction (C).**
  *Premise:* "The museum is closed on Mondays."
  *Hypothesis:* "The museum is open every Monday."
- **Neutral (N).**
  *Premise:* "A chef entered the kitchen."
  *Hypothesis:* "The chef prepared pasta."

**B. Internal contradiction example**

- **Yes (contradictory).**
  *Premise:* "The event starts at 8 pm and starts at 9 pm."
  *Hypothesis:* (any)

**C. Redundancy examples**

- **None.** "The match took place in Paris." (no repetition)
- **Some.** "The match took place in Paris, France." (light paraphrase once)
- **Many.** "The match took place in Paris, which is in France. The game occurred in France, specifically Paris." (repeated content)

**D. Factuality vs. fact list examples**

- **Facts:** F1 "The Eiffel Tower is in Paris."   F2 "Roland Garros is a tennis venue in Paris."
- **Supported.**
  *Premise:* "The final was held at Roland Garros in Paris."
  *Hypothesis:* "The final took place in Paris."
  *Why:* Premise aligns with F2; hypothesis follows from the premise.
- **Partially.**
  *Premise:* "The final was held at a major stadium."
  *Hypothesis:* "The final took place in Paris."
  *Why:* "Major stadium" is underspecified by F1/F2; location remains unclear.
- **Unsupported.**
  *Premise:* "The final was held in Berlin."
  *Hypothesis:* "The final took place in Paris."
  *Why:* Conflicts with the premise and not backed by F1/F2.

**E. Readability anchors (Likert)**

- **1 (unreadable).** "Win match rain delay heavy because." (severe errors)
- **3 (adequate).** "The match was delayed due to rain; wording is a bit awkward but clear."
- **5 (clear).** "Heavy rain delayed the match." (natural and fluent)

## H.2 HUMAN EVALUATION SETUP

**Scope and blinding.** Annotators see only the *enhanced* premise and hypothesis plus a compact *fact list* (3–15 atomic facts) for each item. They do *not* see original/root texts, original labels, model predictions, or the system variant (Full vs. ablation). Item order is individually randomized to avoid order and fatigue effects.

**Sampling.** We draw a stratified sample from the enhanced pool with strata defined by *dataset* (SNLI, MNLI) and *level* $L \in \{1, 2, 3\}$. The sampling frame contains 9,824 SNLI items per level and 19,647 MNLI items per level. Within each $(\text{dataset}, L)$ stratum, we uniformly sample without replacement a target of $n_s$ items—SNLI: $n_s{=}20$ per level; MNLI: $n_s{=}40$ per level. This yields

$$N_{\text{enhanced}} = (3 \times 20)_{\text{SNLI}} + (3 \times 40)_{\text{MNLI}} = 180.$$

Where feasible, we balance the root label distribution (E/N/C) *at sampling time only* to ensure coverage; annotators are not shown root labels.

Separately, from the NO-FILTER ablation sets we sample $L{=}3$ only: 20 SNLI items and 40 MNLI items, i.e.,

$$N_{\text{ablation}} = 20_{\text{SNLI}} + 40_{\text{MNLI}} = 60.$$

Thus the combined total is

$$N_{\text{all}} = N_{\text{enhanced}} + N_{\text{ablation}} = 240.$$

