# OpenReview forum: "FactNLI: Dynamic and Automated Fact-Enhanced Augmentation of NLI Benchmarks"
_ICLR.cc/2026/Conference — ICLR 2026 Conference Desk Rejected Submission_

### Official Review · Reviewer_DJoJ · 2025-10-21

**Soundness:** 2
**Presentation:** 2
**Contribution:** 2
**Rating:** 2
**Confidence:** 4

**Summary:**

This paper proposes a framework for automatically augmenting NLI datasets by iteratively fusing verified facts from Wikipedia into premise-hypothesis pairs. The method uses retrieval, filtering, and controlled fusion to create progressively harder instances while preserving original labels. Experiments on SNLI and MNLI show consistent accuracy drops (up to 30%) across multiple LLMs. Human evaluation was also conducted to show the generated data are of high-quality.

**Strengths:**

- The proposed fact-augmented framework is carefully designed and can be used as a valuable data augmentation method.

**Weaknesses:**

- The motivation for the paper is not sound. The authors framed this work as, because existing benchmarks are saturated, we need a way to dynamically update them. This framing is ok but adding Wikipedia facts to NLI samples seems weird:
  + First, benchmarks need to reflect real-world usage. If a benchmark is saturated by current models, we need to identify what are the remaining challenges in order to construct a new one. Augmenting the samples with arbitrary facts from Wikipedia move the benchmark away from its intended purpose. Are we evaluating fact-checking capabilities or evaluating the inference relationship in NLI task?
  + Looking at the augmented samples, it seems that the method just add irrelevant padding. If the goal is just to make the text longer and overload with more information, we can just ask the LLM to rewrite the samples to make them longer.
  + More importantly, do the retrieved facts actually influent in the inference? The process of fusing and verifying facts is good but does it matter if the facts are not needed to infer the correct label.
- The paper demonstrates difficulty increase but doesn't analyze what makes augmented instances harder, e.g., length, coherence, too many entities, multi-hop reasoning.

**Questions:**

I would be more convinced about the claims if the author could do experiments to:
- Show that facts participate in reasoning.
- Compare with simple expansion baselines, such as simply prompt the LLMs to rewrite the NLI samples to make it longer without changing the label.

---

> ### Author Response · Authors · 2025-11-20
>
> We thank the reviewer for the careful reading and constructive suggestions. We have substantially revised the paper and added an _Analysis of Difficulty_ subsection to address these concerns.
>
> > The motivation for the paper is not sound. The authors framed this work as, because existing benchmarks are saturated, we need a way to dynamically update them. This framing is ok but adding Wikipedia facts to NLI samples seems weird:
> >
> > - First, benchmarks need to reflect real-world usage. If a benchmark is saturated by current models, we need to identify what are the remaining challenges in order to construct a new one. Augmenting the samples with arbitrary facts from Wikipedia moves the benchmark away from its intended purpose. Are we evaluating fact-checking capabilities or evaluating the inference relationship in the NLI task?
> >
>
> Regarding our **motivation**, it is important to emphasize that real-world usage of NLI, especially in tasks like hallucination detection, is inherently complex. In such scenarios, the task is not simply about matching short premise and hypothesis, but rather about inferring through a large set of facts to determine whether they support or contradict each other. This level of inference is often required in real-world applications. We believe that current NLI datasets are overly **simple and static**, failing to capture the complexity of these real-world scenarios. The goal of our work is to dynamically augment the data, reflecting the intricacies of practical NLI tasks. In this context, Wikipedia, as a rich and frequently updated corpus, aligns better with real-world usage. Our task remains firmly rooted in NLI, not generic fact-checking or arbitrary Wikipedia padding, and we have ensured that the augmented facts are relevant and interact meaningfully with the original premise and hypothesis.
>
> > - Looking at the augmented samples, it seems that the method just adds irrelevant padding. If the goal is just to make the text longer and overload it with more information, we can just ask the LLM to rewrite the samples to make them longer.
> >
> > - Compare with simple expansion baselines, such as simply prompting the LLMs to rewrite the NLI samples to make it longer without changing the label.
> >
> > - The paper demonstrates difficulty increase but doesn't analyze what makes augmented instances harder, e.g., length, coherence, too many entities, multi-hop reasoning.
> >
> > - More importantly, do the retrieved facts actually influence the inference? The process of fusing and verifying facts is good, but does it matter if the facts are not needed to infer the correct label?
> >
> > - Show that facts participate in reasoning.
> >
>
>
> |Model|Original|FactNLI $L{=}1$|Rewrite|Unrelated-Facts|
> |---|---|---|---|---|
> |GPT-4o|84.8|62.1|78.6|80.7|
> |DeepSeek-V3|81.3|64.8|76.9|71.9|
> |DeBERTa-v3-large|92.4|77.1|80.4|85.2|
> |Qwen2.5-14B-Instruct|82.4|64.7|78.0|72.1|
>
> Following your suggestion, we added a **rewrite-only baseline** where GPT-4o expands (p,h) to match the FactNLI L=1 length distribution without adding any external facts. This produces only modest drops (e.g., GPT-4o on SNLI: 84.8 → 78.6), whereas FactNLI-L=1 causes much larger drops on the same data (84.8 → 62.1). This shows that increased length alone does not account for the observed difficulty.
>
> We additionally introduce an **unrelated-facts ablation** to **detect whether injected facts actually participate in inference.** Using the same fusion protocol but replacing our aligned Wikipedia facts with length-matched, off-topic statements, accuracy in this condition stays close to the original data (e.g., GPT-4o on SNLI: 84.8 → 80.7), and is much higher than on FactNLI-L=1. Since length and surface form are comparable, the extra difficulty must come from the semantically aligned facts that interact with the entities/events in (p,h), rather than from generic noise or extra tokens.
>
> The experimental data is shown in the table . More details and analysis can be found in the appendix of the updated paper.
>
> Together, these new experiments directly address the reviewer’s questions: (i) we compare against simple expansion baselines, and (ii) we show that it is specifically the label-preserving, aligned facts—not mere length or padding—that participate in inference and make the augmented instances harder.
>
> We sincerely thank the reviewer for the thoughtful and constructive feedback.

---

> > ### Comment · Reviewer_DJoJ · 2025-11-25
> > **Response to authors**
> >
> > Thank you for your efforts in addressing my concerns.
> >
> > Regarding your first point, the goal of NLI is on testing predicate logic, reasoning (even in counterfactual, or hypothetical P/H pairs) and shouldn't be relying on real-world knowledge. Using Wikipedia as a source to augment data inherently moving the task to be more aligned with fact checking (also making the models biased toward their parametric knowledge).
> >
> > The new ablation on Rewrite and Unrelated facts are helpful but their results need to be analyzed more carefully (such as how do you ensure the ground truth are still correct after adding unrelated fact). I understand that given the limited time, it's hard to do this. However, it does show that the FactNLI pipeline does introduce non-trivial perturbation to the models.
> >
> > I have considered my reviews and your response and have adjusted the scores accordingly.

---

> ### Author Response · Authors · 2025-11-27
>
> Dear Reviewer,
>
> Thank you very much for your detailed feedback and thoughtful understanding of our work. We greatly appreciate the time and effort you have dedicated to reviewing our manuscript. We greatly appreciate your adjustment of the scores. But we hope there is still an opportunity to provide some clarification.
>
> > Regarding your first point, the goal of NLI is to test predicate logic and reasoning (even in counterfactual or hypothetical P/H pairs) and should not rely on real-world knowledge. Using Wikipedia as a source to augment data inherently moves the task closer to fact-checking, which could also make the models biased toward their parametric knowledge.
>
> We fully agree with your perspective that the goal of NLI is to test predicate logic and reasoning, and that it should not rely on real-world knowledge. In this context, the augmented task does not require the model to access **external knowledge** when evaluating the premise-hypothesis relationship. For example, when we introduce an atomic fact such as “this is located in Paris,” we add it symmetrically to both the premise and the hypothesis, and deliberately avoid cases like “this is in France” vs. “this is in Paris,” which would require background knowledge that Paris is in France. In fact, the real-world facts introduced by our augmentation framework serve as shared background information within the text, and the model is never required to verify their truth against external knowledge.
>
> > The new ablation on Rewrite and Unrelated facts is helpful, but their results need to be analyzed more carefully (such as how do you ensure the ground truth remains correct after adding unrelated facts). I understand that given the limited time, it's hard to do this. However, it does show that the FactNLI pipeline introduces non-trivial perturbation to the models.
>
> Regarding the rewritten texts, we carefully controlled the quality of the ablation experiment by restricting their length to be comparable to that of the L=1 setting in FactNLI. This is because at greater lengths the model is much more likely to introduce extraneous information. Under the current setup, we performed small-batch spot checks and manual verification, confirming that the rewritten texts remain strictly consistent with the originals in both semantics and facts.
>
> Regarding the introduction of unrelated facts, we select trivial but true statements from a **commonsense** corpus (e.g., “water boils at 100 degrees”) and insert them into the pairs. These facts are independent of the core premise–hypothesis relationship and do not introduce any logical or semantic links that could change the label; they serve only as label-irrelevant background noise, without affecting the core reasoning task.
>
> Thank you once again for your valuable comments. We believe the clarifications provided address your concerns and hope they offer a clearer understanding of how our augmentation framework preserves the integrity of the NLI task. We truly appreciate your thoughtful review and remain open to any further discussion or suggestions.

---

### Official Review · Reviewer_ta4B · 2025-10-23

**Soundness:** 3
**Presentation:** 4
**Contribution:** 3
**Rating:** 8
**Confidence:** 3

**Summary:**

This paper addresses a well-recognized and critical problem in NLP: the saturation of existing NLI benchmarks like SNLI and MNLI by modern LLMs. The authors argue that the simplicity and static nature of these datasets fail to truly challenge current models and reveal their shortcomings. To this end, they propose FACTNLI, a novel and well-designed framework for automatically augmenting existing NLI instances with verifiable facts from Wikipedia. The core idea is to iteratively retrieve, filter, and fuse external facts into the premise and hypothesis, thereby increasing their length, semantic complexity, and the required reasoning depth, all while preserving the original inference label.

**Strengths:**

1. The problem addressed is timely and of great importance. As LLMs continue to improve, the need for more challenging and dynamic evaluation benchmarks is paramount. This work offers a scalable and principled solution, moving beyond static, one-off dataset collection efforts.
2. The proposed retrieve-filter-fuse pipeline is well-motivated and technically sound. It provides a robust mechanism to ensure logical consistency and prevent the introduction of contradictory or redundant information, which is a common pitfall of automated data generation. Grounding the augmentation in an external knowledge source (Wikipedia) adds a layer of verifiability that is often missing in purely synthetic generation approaches.
3. The authors evaluate a diverse set of seven models, spanning different architectures and sizes, which demonstrates the general applicability of their findings. And the multi-level augmentation provides a clear, controllable knob for difficulty, and the results compellingly show a monotonic decrease in performance as complexity increases.
4. The paper is well-written and easy to follow. Figure 1 provides an excellent intuitive example, and Figure 2 clearly illustrates the overall framework.

**Weaknesses:**

1. The current implementation totally relies on Wikipedia. While Wikipedia is comprehensive, this limits the framework's applicability to specific domains not well-covered by it (e.g., specialized medical or legal texts). A discussion of how the framework could be adapted to other knowledge sources or specific domains would strengthen the paper.
2. The fusion step uses GPT-4o to compose the new premise and hypothesis. This process may introduce stylistic biases or artifacts characteristic of the generator model. It is possible that models evaluated on this data could learn to exploit these stylistic cues. While the authors separate generation from filtering, a brief analysis of the linguistic style of the augmented text (e.g., using a classifier to distinguish it from human-written text) would be a valuable addition.
3. The paper demonstrates that models fail more often on the augmented data, but a deeper analysis of why they fail would be more insightful. Are the failures primarily due to the increased context length, or do they stem from an inability to perform multi-fact reasoning? A qualitative or quantitative error analysis categorizing the types of reasoning required (e.g., temporal, spatial, numerical) and where models struggle most would provide a richer understanding of the challenges introduced.

**Questions:**

1. The framework involves many calls to GPT-4o and other models for each sample at each level. Can you provide a brief comment on the computational resources and time required to generate the augmented datasets? This would be helpful for other researchers looking to apply this method.
2. Have you considered the effect of the retrieved facts' quality? The retrieval step is based on semantic similarity. Is it possible that loosely related or slightly inaccurate snippets are retrieved, and how does the downstream filtering and extraction pipeline handle such noise?

---

> ### Author Response · Authors · 2025-11-20
>
> We thank the reviewer for the careful reading and constructive suggestions. We have substantially revised the paper and added _Analysis of Difficulty_, _Open Source Generation_ subsection in Appendix to address these concerns.
>
> > The current implementation totally relies on Wikipedia. While Wikipedia is comprehensive, this limits the framework's applicability to specific domains not well-covered by it (e.g., specialized medical or legal texts). A discussion of how the framework could be adapted to other knowledge sources or specific domains would strengthen the paper.
>
> In response to the reliance on Wikipedia, we clarify that considering SNLI and MNLI datasets are highly general and cover a wide range of domains, Wikipedia is used as a convenient instantiation of the external knowledge source rather than a fundamental requirement. The framework itself is source-agnostic. In the future, for more domain-specific NLI tasks, we could apply our framework to use specialized corpora relevant to the target domain, ensuring that the facts injected are aligned with the specific terminology and knowledge of the domain.
>
> > The paper demonstrates that models fail more often on the augmented data, but a deeper analysis of why they fail would be more insightful. Are the failures primarily due to the increased context length, or do they stem from an inability to perform multi-fact reasoning? A qualitative or quantitative error analysis categorizing the types of reasoning required (e.g., temporal, spatial, numerical) and where models struggle most would provide a richer understanding of the challenges introduced.
>
> |Model|Original|FactNLI $L{=}1$|Rewrite|Unrelated-Facts|
> |---|---|---|---|---|
> |GPT-4o|84.8|62.1|78.6|80.7|
> |DeepSeek-V3|81.3|64.8|76.9|71.9|
> |DeBERTa-v3-large|92.4|77.1|80.4|85.2|
> |Qwen2.5-14B-Instruct|82.4|64.7|78.0|72.1|
>
> To analyze why the difficulty increases, we added a **rewrite-only baseline** where GPT-4o expands (p,h) to match the FactNLI L=1 length distribution without adding any external facts. This produces only modest drops (e.g., GPT-4o on SNLI: 84.8 → 78.6), whereas FactNLI-L=1 causes much larger drops on the same data (84.8 → 62.1). This shows that increased length alone does not account for the observed difficulty.
>
> We additionally introduce an **unrelated-facts ablation** to detect whether injected facts actually participate in **inference** or just distract models. Using the same fusion protocol but replacing our aligned Wikipedia facts with length-matched, off-topic statements, accuracy in this condition stays close to the original data (e.g., GPT-4o on SNLI: 84.8 → 80.7), and is much higher than on FactNLI-L=1. Since length and surface form are comparable, the extra difficulty must come from the semantically aligned facts that interact with the entities/events in (p,h), rather than from generic noise or extra tokens.
>
> The experimental data is shown in the table above. More details and analysis can be found in the Appendix _Analysis of Difficulty_ of the updated paper.
>
> To analyze the **error patterns**, we should first distinguish between three labels. Entailment means that the premise can directly infer the hypothesis. Neutral means that the hypothesis contains information not present in the premise, which could be either correct or incorrect. Contradiction means that under the premise, the hypothesis cannot possibly occur. Since the introduced facts often contain multiple pieces of information, it is difficult to analyze specifically why a particular inference fails. However, it is clear that the errors are consistent: the model can detect the differences between the premise and hypothesis information, but it struggles to distinguish the likelihood of such differences occurring. This type of error can also be understood as a core challenge in NLI tasks.

---

> ### Author Response · Authors · 2025-11-20
>
> > The fusion step uses GPT-4o to compose the new premise and hypothesis. This process may introduce stylistic biases or artifacts characteristic of the generator model. It is possible that models evaluated on this data could learn to exploit these stylistic cues. While the authors separate generation from filtering, a brief analysis of the linguistic style of the augmented text (e.g., using a classifier to distinguish it from human-written text) would be a valuable addition.
>
> If the reviewer is referring to model bias, note that GPT-4o does not participate in any NLI-dependent judgment within the framework; the generation process is rule-based and does not introduce model bias in NLI tasks. If the reviewer is concerned that the linguistic style of GPT-4o-generated text could influence the model's judgment, it’s important to emphasize the consistency of the augmentation methods for the three labels under label preservation. The language style would not mislead the prediction results. If we consider whether this linguistic style affects overall understanding, we observe that in the difficulty analysis experiments, the model maintains high accuracy on GPT-4o-generated rewrite samples, so the impact of this style should be minimal. The readability assessment during the manual inspection phase also confirms the high quality of the generated data.
>
> > The framework involves many calls to GPT-4o and other models for each sample at each level. Can you provide a brief comment on the computational resources and time required to generate the augmented datasets? This would be helpful for other researchers looking to apply this method.
>
> The framework indeed involves multiple GPT-4o calls per sample and level. We estimated the average input and output token counts. Generating one set of (level1, level2, level3) samples requires approximately 4k input tokens and 1k output tokens. The cost of generating 10,000 sets using GPT-4o is about $200. The generation speed of a single sample depends on the API call speed, which takes around 20 seconds. For large-scale sample generation, the time mainly depends on the number of parallel threads. With 20 parallel threads, it takes about 3 hours to generate 10,000 sets.
>
> > Have you considered the effect of the retrieved facts' quality? The retrieval step is based on semantic similarity. Is it possible that loosely related or slightly inaccurate snippets are retrieved, and how does the downstream filtering and extraction pipeline handle such noise?
>
> In fact, the noise in this task might be understood differently compared to the reviewer’s interpretation: **whether the facts are strictly accurate doesn’t matter**. The key aspect of NLI tasks is whether the hypothesis can be inferred from the premise. Therefore, slightly inaccurate facts will not affect the inference of the NLI samples. As for **loosely related facts**, this is actually one of the design requirements. Highly relevant facts would be semantically redundant, and unrelated topics have already been filtered out during the retrieval stage. What we introduce are facts that have a certain degree of relevance and can seamlessly integrate with the original sample to form a coherent, enhanced example. For data augmentation, **the truly harmful noise** is any fact that conflicts with the original example, which will affect label preservation. These types of noise are already removed during the filtering stage. Moreover, human annotation results also confirm that our mechanism ensures the final dataset is high-quality across multiple dimensions.
>
> We hope these clarifications and additions address the reviewer’s concerns about domain generality, stylistic artifacts, failure analysis, resource requirements, and retrieval noise.We sincerely thank the reviewer for the thoughtful and constructive feedback.

---

> > ### Comment · Reviewer_ta4B · 2025-11-26
> >
> > I thank the authors for their detailed response and for conducting the additional ablation studies. These experiments effectively clarify that the performance drop is driven by the injected logic rather than mere sequence length, which addresses one of my major concerns. I believe my original score accurately reflects the paper's strengths and its remaining weaknesses. Therefore, I will maintain my current rating.

---

### Official Review · Reviewer_Yg7r · 2025-11-02

**Soundness:** 3
**Presentation:** 4
**Contribution:** 3
**Rating:** 6
**Confidence:** 4

**Summary:**

The paper argues that traditional NLI benchmarks are no longer challenging leading to iterative development of more challenging benchmarks through (1) adversarial approaches or (2) domain-specific benchmark development. However, both these approaches are time intensive and require careful human annotation. Instead, this paper proposes an automated pipeline to generate challenging benchmarks that can be made more successively more difficult when NLI methods saturate the previous iteration.

Stage 0 corresponds to the existing benchmarks (SNLI and MNLI in this paper). Each successive stage adds meaning preserving facts to the premise and hypothesis pair from the previous stage such that the entailment label does not change. Results show that accuracy of models, achieving 80-90% in stage 0, drop by 10-20% with just one iteration of the FactNLI process (length of premises increases 3.5x on average).

FactNLI increases difficulty of the task by adding relevant but distracting factual information to the premise in hypothesis. It applies the following steps to each (premise, hypothesis, label) triplet in the previous iteration of the benchmark.
1. Retrieve introductory passages from Wikipedia based on the premise (using sentence embedding models)
2. Extract atomic facts from the passages (using GPT-4o)
3. Filter our atomic facts that are not neutral with respect to the premise (NLI judged by a small fine-tuned model based on DeBERTa-V3-Base)
4. Keep the maximal clique of facts that are each pairwise neutral with each other (NLI judged by a small fine-tuned model based on DeBERTa-V3-Base)
5. Fuse the clique of facts into the premise from the previous stage. Fuse a **subset** of facts from each iteration into the **original** (stage 0) hypothesis. The label is retained as is (using GPT-4o)

Assumptions: GPT-4o can perform atomic fact decomposition and fact fusion without introducing errors. The small NLI model is accurate for judging single sentence hypotheses.

The quality of the benchmark is validated across several axes (most importantly label accuracy, fusion errors, introduced logical inconsistencies) with a stratified human annotation (60 examples per stage x 3 stages of the final benchmark). Results show that the label is preserved 95% of the time after 3 stages with less than 7% of examples with internal inconsistency.

**Strengths:**

1. The method is demonstrated to cause iterative decrease in accuracy of NLI classifiers
2. The quality degradation is studied in detail using human annotations
3. Ablations show the effectiveness of the filtering stage

**Weaknesses:**

1. (Not a reason to reject) The difficulty introduced is one-dimensional and this method results in a sanity check benchmark (akin to Needle-in-a-haystack for long-context LLMs and Checklists in past NLI benchmarks)
    - High performance on the benchmark demonstrates the ability of NLI models to handle long noisy context where *most* atomic facts in the premise and hypothesis agree by construction
2. (Necessary validation to characterize the difficulty) I believe that by design, FactNLI will induce more queries to be predicted as entailed by NLI models. This is because only one atomic fact differs between the contradictory premise and hypothesis. If models cannot detect this one misaligned fact, they are more likely to predict that the entailed label.
    - A confusion matrix of the errors in each label category can help clarify my hunch
    - If the authors disagree, can you characterize the errors made by the NLI models in Stages 1, 2, and 3?
3. (Unclear behavior of neutral examples) It is unclear how the neutral label is maintained after the fact fusion. Can the authors provide examples from the benchmark to explain how neutrality is maintained?
4. (Missing discussion) Ambiguity is a prevailing issue for NLI models and this discussion is missing in the paper. E.g. AmbiEnt (Liu et al, EMNLP 2023) show that models struggle to capture the ambiguity in human labelers and cannot successfully disambiguate entailment queries with ambiguity.
    - This is also relevant to the sample discussed in the Appendix D. It can be argued that L1 Hypothesis is no longer entailed by the L1 Premise, because L1 Premise associates the street fashion with the city street while the hypothesis associates the fashion to the woman.
    - The human annotation is thus necessary to show that such effects are not introduced very frequently

**Questions:**

Questions are asked in the previous section.

---

> ### Author Response · Authors · 2025-11-20
>
> We thank the reviewer for the careful reading and constructive suggestions. We have substantially revised the paper and added _Analysis of Difficulty_, _Confusion Matrices_ subsection in Appendix to address these concerns.
>
> > (Not a reason to reject) The difficulty introduced is one-dimensional and this method results in a sanity check benchmark (akin to Needle-in-a-haystack for long-context LLMs and Checklists in past NLI benchmarks)
>
> - High performance on the benchmark demonstrates the ability of NLI models to handle long noisy context where _most_ atomic facts in the premise and hypothesis agree by construction
>
>
> |Model|Original|FactNLI $L{=}1$|Rewrite|Unrelated-Facts|
> |---|---|---|---|---|
> |GPT-4o|84.8|62.1|78.6|80.7|
> |DeepSeek-V3|81.3|64.8|76.9|71.9|
> |DeBERTa-v3-large|92.4|77.1|80.4|85.2|
> |Qwen2.5-14B-Instruct|82.4|64.7|78.0|72.1|
>
> In response to the concern regarding the one-dimensional difficulty increase, we conducted a difficulty analysis. We added a **rewrite-only baseline** where GPT-4o expands (p,h) to match the FactNLI L=1 length distribution without adding any external facts. We also introduced an **unrelated-facts ablation** to detect whether injected facts actually participate in **inference** or just distract models. Using the same fusion protocol but replacing our aligned Wikipedia facts with length-matched, off-topic statements, accuracy in both conditions stay close to the original data and are much higher than on FactNLI-L=1. Since length and surface form are comparable, the difficulty increase does not solely arise from the "Needle-in-a-haystack" challenge for long-context LLMs but rather from the need to incorporate facts into deeper inference. The experimental data is shown in the table below. More details and analysis can be found in the appendix of the updated paper.
>
> > (Necessary validation to characterize the difficulty) I believe that by design, FactNLI will induce more queries to be predicted as entailed by NLI models. This is because only one atomic fact differs between the contradictory premise and hypothesis. If models cannot detect this one misaligned fact, they are more likely to predict that the entailed label.
>
> |Gold \ Pred|E|N|C|
> |---|---|---|---|
> |**E**|75.0%|20.7%|4.3%|
> |**N**|30.5%|56.2%|13.3%|
> |**C**|2.8%|35.3%|61.9%|
>
> We added label-wise **confusion matrices** in Appendix Confusion Matrices. They do not support a collapse toward entailment. Contradictions are much more often misclassified as neutral than as entailment, and all three labels see reduced diagonal mass as we move from L=0 to higher levels. This indicates that models are not simply over-predicting entailment; rather, they frequently fail to locate which of many aligned facts is decisive and hedge toward neutral or otherwise mislabel. An average Level1 confusion matrix is shown above.
>
> To analyze the error patterns, we should first distinguish between neutral and contradiction. Neutral means that the hypothesis contains information not present in the premise, which could be either correct or incorrect. Contradiction means that under the premise, the hypothesis cannot possibly occur. Our analysis suggests that for augmented samples, models are often able to identify the informational differences between the premise and hypothesis. However, determining whether this difference suggests a neutral relationship, a likely entailment, or an impossible contradiction requires stronger reasoning. As a result, models often mistake neutral as entailment and contradiction as neutral.

---

> ### Author Response · Authors · 2025-11-20
>
> > (Unclear behavior of neutral examples) It is unclear how the neutral label is maintained after the fact fusion. Can the authors provide examples from the benchmark to explain how neutrality is maintained?
>
> Since **Neutral** means that the hypothesis contains information not present in the premise, which could be either correct or incorrect, it can be concluded that as long as the added facts during augmentation do not allow the hypothesis to be entirely inferred from the premise, the neutral label will be preserved. Filtered facts ensure this. A Level 1 neutral case is shown below.
>
> **Root premise:** People jump over a mountain crevasse on a rope.
> **Root hypothesis:** Some people look visually afraid to jump.
> **Premise1:** People jump over a mountain crevasse on a rope, engaging in mountaineering-related activities such as traditional outdoor climbing, skiing, and traversing via ferratas. Locations like Grouse Mountain Resort, an alpine ski area with a maximum elevation of over 1,200 m at its peak, offer additional opportunities for these thrilling pursuits.
> **Hypothesis1:** Some people look visually afraid to jump in mountaineering-related activities such as skiing at Grouse Mountain Resort, which has a maximum elevation of over 1,200 m at its peak.
> **Analysis:** The hypothesis "some people look visually afraid to jump" may be true in certain situations, but the premise does not explicitly state whether people are afraid to jump.
>
> > (Missing discussion) Ambiguity is a prevailing issue for NLI models, and this discussion is missing in the paper. E.g., AmbiEnt (Liu et al., EMNLP 2023) show that models struggle to capture the ambiguity in human labelers and cannot successfully disambiguate entailment queries with ambiguity.
>
> In response to ambiguity, upon closer inspection, we find that the few ambiguous cases mostly arise because the fusion model occasionally uses inappropriate logical connectives when stitching together new facts, which can inadvertently suggest stronger logical relations than intended. However, downstream NLI models do not appear to rely systematically on these connectives for label decisions: for example, in the street-fashion case in Appendix D, the model still correctly predicts entailment. In a random sample of 50 augmented instances, only 2 were judged to have their gold label potentially affected by connective-induced logical relations. We further experimented with a revised fusion prompt that explicitly instructs the model not to introduce unnecessary logical connectives between newly added facts, and found that this largely eliminates such rare issues.
>
> We hope these additions clarify the difficulty profile of FactNLI and address your questions about label-wise behavior, neutrality, and ambiguity.We sincerely thank the reviewer for the thoughtful and constructive feedback.

---

> ### Author Response · Authors · 2025-11-27
>
> Dear Reviewer,
>
> I hope this message finds you well. I would like to kindly follow up regarding the review of our manuscript. We greatly appreciate the time and effort you have dedicated to evaluating our work, and we truly value your thoughtful insights. In response to your feedback, we have conducted additional experiments, updated the manuscript, and made the necessary clarifications. We look forward to hearing your thoughts on these updates.
>
> Please let us know if there is any additional information we can provide to assist you in the process. Thank you once again for your time and consideration. We look forward to hearing from you.

---

### Official Review · Reviewer_6CCZ · 2025-11-04

**Soundness:** 3
**Presentation:** 2
**Contribution:** 2
**Rating:** 4
**Confidence:** 3

**Summary:**

This paper presents FactNLI, an automated framework designed to augment existing NLI benchmarks like SNLI and MNLI. The authors argue that current datasets are too simple and static for modern LLMs. The proposed method iteratively retrieves verifiable facts from Wikipedia and fuses them into the original premise and hypothesis pairs. This process is structured in multiple levels to progressively increase difficulty. A key component is a filtering pipeline that uses a truth set and an entailment graph to ensure fused facts are non contradictory and non redundant, thereby preserving the original NLI label. Experiments show this augmentation significantly increases text complexity and leads to large performance drops for several LLMs. The authors support their method with a human audit that confirms the high quality and label fidelity of the augmented data.

**Strengths:**

1. The work addresses a well known and important problem. Static benchmarks like SNLI and MNLI are saturated, and model performance on them is no longer a reliable indicator of true language understanding.
2. The proposed framework is systematic and well designed. The idea of using a multi level augmentation to control difficulty is sensible.
3. The inclusion of a truth set and graph based filtering mechanism is a crucial strength. It directly tackles the most common failure mode of data augmentation, which is label drift and the introduction of artifacts.

**Weaknesses:**

1. The primary weakness is the contribution's novelty. The framework is essentially a very well executed data generation pipeline. It relies heavily on an existing proprietary model (GPT-4o) for the core tasks of fact extraction and fusion, and on standard NLI models for filtering. This feels more like a strong engineering contribution for dataset creation rather than a novel method for NLI.
2. The paper does not fully disentangle why the task becomes harder. The augmented examples are up to 10x longer. It is unclear if the performance drop is due to a failure in complex logical reasoning or simply a failure in long context processing and aggregation of (sometimes distracting) facts. The task may be testing retrieval and summarization more than inference.
3. The framework's goal of strict label preservation seems like a missed opportunity. In a dynamic, fact-centric setting, adding new evidence could and perhaps should logically change the relationship. The current design seems to force the original label, which may not be the most realistic way to test evidence based reasoning.

Minor:
1. In Figure 1, "great infulence" should be "great influence".
2. In Figure 2, there are several misspellings such as "Retrival" (Retrieval) and "infulence" (influence).
3. In Section 4.5, the heading "Reults and Analysis" should be "Results and Analysis".

**Questions:**

1.The performance drop is significant, but is this because the models must perform deeper inference, or are they simply getting "distracted" by the large volume of added facts, which are sometimes only tangentially related to the original premise? How could you separate these two failure modes?

2.Given the heavy reliance on GPT-4o, did you experiment with any open source models for the fact extraction and fusion steps? How much does the quality of the augmentation, particularly label preservation and coherence, degrade when using smaller models?

---

> ### Author Response · Authors · 2025-11-20
>
> We thank the reviewer for the careful reading and constructive suggestions. We have substantially revised the paper and added _Analysis of Difficulty_, _Open Source Generation_ subsection in Appendix to address these concerns.
>
> > The primary weakness is the contribution's novelty. The framework is essentially a very well-executed data generation pipeline. It relies heavily on an existing proprietary model (GPT-4o) for the core tasks of fact extraction and fusion, and on standard NLI models for filtering. This feels more like a strong engineering contribution for dataset creation rather than a novel method for NLI.
>
> We would like to clarify that the novelty of our work lies in the **dynamic generation of NLI data with controllable difficulty** through the injection of relevant factual information. Rather than relying on rewriting, paraphrasing, or manual relabeling, our approach deliberately uses curated factual content to increase the **reasoning complexity** of the task while preserving the original semantics. The core idea is our rule-based framework for fact selection and filtering, which ensures label consistency and factual relevance, enabling the construction of more realistic and inference-challenging data. The use of GPT-4o is simply an **implementation choice** for fact extraction and fusion—its role is constrained and does not participate in any NLI decision-making—which further distinguishes our framework from previous augmentation paradigms.
>
> > The paper does not fully disentangle why the task becomes harder. The augmented examples are up to 10x longer. It is unclear if the performance drop is due to a failure in complex logical reasoning or simply a failure in long-context processing and aggregation of (sometimes distracting) facts. The task may be testing retrieval and summarization more than inference.
> > The performance drop is significant, but is this because the models must perform deeper inference, or are they simply getting "distracted" by the large volume of added facts, which are sometimes only tangentially related to the original premise? How could you separate these two failure modes?
>
> | Model                | Original | FactNLI $L{=}1$ | Rewrite | Unrelated-Facts |
> | -------------------- | -------- | --------------- | ------- | --------------- |
> | GPT-4o               | 84.8     | 62.1            | 78.6    | 80.7            |
> | DeepSeek-V3          | 81.3     | 64.8            | 76.9    | 71.9            |
> | DeBERTa-v3-large     | 92.4     | 77.1            | 80.4    | 85.2            |
> | Qwen2.5-14B-Instruct | 82.4     | 64.7            | 78.0    | 72.1            |
>
> To analyze why the difficulty increases, we add a **rewrite-only baseline** where GPT-4o expands (p,h) to match the FactNLI L=1 length distribution without adding any external facts. This produces only modest drops (e.g., GPT-4o on SNLI: 84.8 → 78.6), whereas FactNLI-L=1 causes much larger drops on the same data (84.8 → 62.1). This shows that increased length alone does not account for the observed difficulty.
>
> We additionally introduce an **unrelated-facts ablation** to detect whether injected facts actually participate in **inference** or just distract models. Using the same fusion protocol but replacing our aligned Wikipedia facts with length-matched, off-topic statements, accuracy in this condition stays close to the original data (e.g., GPT-4o on SNLI: 84.8 → 80.7), and is much higher than on FactNLI-L=1. Since length and surface form are comparable, the extra difficulty must come from the semantically aligned facts that interact with the entities/events in (p,h), rather than from generic noise or extra tokens.
>
> The experimental data is shown in the table above. More details and analysis can be found in the appendix of the updated paper.
>
> > Given the heavy reliance on GPT-4o, did you experiment with any open source models for the fact extraction and fusion steps? How much does the quality of the augmentation, particularly label preservation and coherence, degrade when using smaller models?
>
> |Model|Label pres. (%)|Conflict (%)|Redundancy (%)|Factuality (%)|Readability|
> |---|---|---|---|---|---|
> |GPT|96.7|4.5|11.7|99.4|4.0|
> |Qwen|93.3|5.0|5.3|91.0|3.5|
>
> In response to the concern regarding our reliance on GPT-4o, we instantiate the generative components with the open-source Qwen2.5-14B-Instruct model on the SNLI dataset. While the open-source model shows a slight decrease in label preservation, factual coverage, and readability compared to the proprietary model, it still maintains an acceptable level of performance. These results suggest that the smaller model is **useable** but does lead to some performance degradation. For cost-effectiveness, using the smaller model is a viable option, but for optimal results, we recommend the use of the larger, proprietary model. This demonstrates that our retrieve-filter-fuse framework is effective across different models.

---

> ### Author Response · Authors · 2025-11-20
>
> > The framework's goal of strict label preservation seems like a missed opportunity. In a dynamic, fact-centric setting, adding new evidence could and perhaps should logically change the relationship. The current design seems to force the original label, which may not be the most realistic way to test evidence-based reasoning.
>
> We understand the concern about strict label preservation. However, we believe that changing labels based on newly added evidence would require additional annotation, which is beyond the scope of our fully **automated framework**. Our approach prioritizes strict label preservation under filtered facts to ensure that the augmentations remain logically consistent and of high quality. This is achieved through a **rule-based system**, which allows us to maintain label consistency automatically. Furthermore, the **manual audit** results, which show high label preservation rates, demonstrate that our approach is effective.
>
> We have corrected the mistakes noted by the reviewer and we sincerely thank the reviewer for the thoughtful and constructive feedback.

---

> ### Author Response · Authors · 2025-11-27
>
> Dear Reviewer,
>
> I hope this message finds you well. I would like to kindly follow up regarding the review of our manuscript. We greatly appreciate the time and effort you have dedicated to evaluating our work, and we truly value your thoughtful insights. In response to your feedback, we have conducted additional experiments, updated the manuscript, and made the necessary clarifications. We look forward to hearing your thoughts on these updates.
>
> Please let us know if there is any additional information we can provide to assist you in the process. Thank you once again for your time and consideration. We look forward to hearing from you.

---

### Comment · Area_Chair_zBA4 · 2025-11-24
**Author Responses Are Ready - Please Review & Provide Feedback**

Dear Reviewers,

Thank you once again for your essential contributions to the review process. The authors have submitted their responses to your initial reviews.

I kindly ask you to carefully review the authors' responses for the papers you are handling. Your timely assessment of how the authors have addressed your original concerns is a critical step in reaching a final decision.

Please provide your feedback and any necessary updates to your reviews as soon as possible to ensure we can meet our tight schedule for the discussion phase.

Your prompt attention to this matter is highly appreciated.

Best regards,

Area Chair

---

### Author Response · Authors · 2025-12-02
**Rebuttal summary**

Dear Area Chair,

This note summarizes how we have addressed the principal weaknesses identified in the reviews—most notably the shared concern regarding **difficulty analysis**—and how these revisions are reflected in the reviewers’ updated reactions.

All four reviewers, and especially Reviewer 6CCZ, Reviewer ta4B, and Reviewer DJoJ, requested a clearer and more quantitative characterization of what makes our FactNLI levels harder. In response, we substantially revised the paper and added an explicit **“Analysis of difficulty”** section with controlled experiments comparing performance on the original data, on FactNLI at L=1, on a rewrite-only variant that matches length and style without adding new facts, and on a variant that only injects unrelated facts. These ablations show that the performance drop cannot be explained by input length or information load alone, but is closely tied to the injected logic. Reviewer ta4B and Reviewer DJoJ explicitly stated that this new analysis resolves their main concern about difficulty, and **Reviewer DJoJ subsequently raised the overall score** for our submission.

Beyond difficulty, we added **label-wise confusion matrices and an error analysis**, which show no collapse toward a single label but systematic struggles in identifying the decisive fact among many. We added **open-source model generation and evaluation** with Qwen2.5-14B alongside GPT-4o, demonstrating that our findings are robust across different generators. We also included a concise **cost analysis** to clarify the practicality and scalability of constructing FactNLI, and modestly expanded the discussion of task **motivation** and the **novelty** of our automatic framework to more clearly situate FactNLI within NLI research without changing the core claims of the paper.

Given that the main cross-reviewer weakness has been acknowledged as resolved in the reviewers’ comments, and that we have systematically addressed the other points with additional experiments and analyses, we respectfully ask that you take these clarifications and the updated reviewer reactions into account when assessing our submission.

---

### Note · Program_Chairs · 2026-01-17
**Submission Desk Rejected by Program Chairs**

The following references in this submission do not refer to real documents and/or have major errors in bibliographic information:

 Hangfeng He, Zhe Zeng, and Heng Ji. Generate, annotate, learn: Generative data augmentation for knowledge-intensive nlp tasks. Transactions of the Association for Computational Linguistics (TACL), 10:154-172, 2022. doi: 10.1162/tacl_a_00449. URL https://aclanthology. org/2022.tacl-1.9/